# Mathematical modelling of bicarbonate supplementation and acid-base chemistry in kidney failure patients on hemodialysis

**Mauro Pietribiasi**◯*, Jacek Waniewski, John K. Leypoldt

Nalecz Institute of Biocybernetics and Biomedical Engineering Polish Academy of Sciences, Warsaw, Poland

* mpietribiasi@ibib.waw.pl

## Abstract

Acid-base regulation by the kidneys is largely missing in end-stage renal disease patients undergoing hemodialysis (HD). Bicarbonate is added to the dialysis fluid during HD to replenish the buffers in the body and neutralize interdialytic acid accumulation. Predicting HD outcomes with mathematical models can help select the optimal patient-specific dialysate composition, but the kinetics of bicarbonate are difficult to quantify, because of the many factors involved in the regulation of the bicarbonate buffer in bodily fluids. We implemented a mathematical model of dissolved $CO_2$ and bicarbonate transport that describes the changes in acid-base equilibrium induced by HD to assess the kinetics of bicarbonate, dissolved $CO_2$, and other buffers not only in plasma but also in erythrocytes, interstitial fluid, and tissue cells; the model also includes respiratory control over the partial pressures of $CO_2$ and oxygen. Clinical data were used to fit the model and identify missing parameters used in theoretical simulations. Our results demonstrate the feasibility of the model in describing the changes to acid-base homeostasis typical of HD, and highlight the importance of respiratory regulation during HD.

## Introduction

Approximately 1 millimole of net endogenous acid per kg of body weight is produced each day by human adults with normal kidney function due primarily to the metabolism of dietary protein [1]. Under normal physiological conditions, the body achieves acid-base balance through the action of the lungs, which exhale carbon dioxide, and the kidneys. When kidney function deteriorates, metabolic acidosis can develop due to the inability of the kidneys to excrete the hydrogen ions produced by endogenous acid production, potentially leading to deleterious catabolic, pro-inflammatory and bone-reabsorbing effects [2].

In end-stage chronic kidney disease, when patients require hemodialysis (HD) therapy, metabolic acidosis is treated by including bicarbonate (and/or other buffer bases such as lactate, citrate, or acetate) to the dialysis fluid, to neutralize net acid production. As a result of the intermittent nature of HD treatments, HD patients are frequently acidotic pre-dialysis but can be alkalotic during and immediately after a treatment. In a recent study, only 4 out of 53 prevalent HD patients had all acid-base blood chemistry pre-dialytic values in the normal range [3].

◯ OPEN ACCESS

**Data Availability Statement:** The data used in this work were previously published in: Sargent JA, Marano M, Marano S, Gennari FJ. Acid-base homeostasis during hemodialysis: New insights

into the mystery of bicarbonate disappearance during treatment. Semin Dial. 2018 Sep;31(5):468-478. doi: 10.1111/sdi.12714. Epub 2018 May 29. PMID: 29813184. Park S, Paredes W, Custodio M, Goel N, Sapkota D, Bandla A, Lynn RI, Reddy SM, Hostetter TH, Abramowitz MK. Intradialytic acid-base changes and organic anion production during high versus low bicarbonate hemodialysis. Am J Physiol Renal Physiol. 2020 Jun 1;318(6):F1418-F1429. doi: 10.1152/ajprenal.00036.2020. Epub 2020 Apr 20. PMID: 32308019; PMCID: PMC7311706.

**Funding:** This work was funded by the Polish National Science Center (grant number: 2017/27/B/ST7/03029). JKL was the beneficiary of the grant. MP was employed in the grant. The funders had no role in study design, data collection and analysis, decision to publish, or preparation of the manuscript.

**Competing interests:** The authors have declared that no competing interests exist.

The prescription of the appropriate dose of basic buffers in dialysis fluid is a difficult task. Observational studies have shown that both high and low pre-dialytic bicarbonate levels are associated with an increased risk of mortality in HD patients [4–7]. Although the prescription of high bicarbonate in dialysis fluid has been recently debated [8,9], KDOQI guidelines recommend that pre-dialytic serum bicarbonate be $\geq$22 mmol/L. Nonetheless, a large fraction of HD patients do not achieve the recommended levels (only 60% of HD patients in the United States do so [10]). There are no quantitative guidelines on how to reach recommended serum bicarbonate levels [11,12].

Several authors have suggested that more in-depth computational studies can assist in the choice of the buffer base composition of the dialysis fluid [9,13]. However, mathematical modeling of acid-base chemistry provides specific challenges; the kinetics of bicarbonate and dissolved carbon dioxide ($CO_2$) during HD are more difficult to describe than that of uremic toxins such as urea or creatinine. Acid-base chemistry in body fluids is complex, involving interactions between bicarbonate, non-bicarbonate buffers and dissolved $CO_2$ in plasma, and the reactions of $CO_2$ to form carbamylated proteins, specifically hemoglobin within the red blood cell [14]. A full description of acid-base kinetics requires also taking into consideration the role of respiration in controlling $CO_2$ levels in blood. The kinetics of bicarbonate transport during HD are themselves not entirely understood, as there is still no consensus on the cause of the observed lack of equilibration between the bicarbonate concentrations in plasma and dialysate, regardless of session length [15–17]. Transport of dissolved $CO_2$ alongside bicarbonate has also been reported to be possible cause of acidosis in respiratory-impaired patients, although it is often neglected [18].

Extensive physicochemical models of acid-base chemistry in intracellular and extracellular fluids have been proposed with or without regulation by pulmonary mechanisms [19–21] but were mostly applied to the study of the physiology of general population, often considering only steady-state conditions. Models specifically describing the effects of the HD treatment have also been proposed, albeit with different degrees of sophistication. Thews and Hutten were the first to implement a complex model (24-compartments) to describe the changes in serum bicarbonate, $CO_2$ and $H^+$ ions during HD, including chemical reactions and respiratory control mechanisms [22]. That model was able to accurately predict the immediate post-dialytic arterial serum bicarbonate and $H^+$ concentrations, but it was too complex for patient-specific implementation, and its parameters were incompletely described [23]. Ursino, Coli, and colleagues modeled the kinetics of bicarbonate and $H^+$ ions during HD alongside other small solutes (sodium, potassium, chloride), but their model neglected the role of lung function, thereby assuming equal arterial and venous concentrations of all solutes [24,25]. Recently, Sargent et al [26] and Wolf [19] proposed simpler compartmental models describing bicarbonate kinetics during HD to explain the lack of equilibration between bicarbonate levels in plasma and dialysis fluid by the end of an HD treatment; these models also neglected lung function and were limited to predictions of intradialytic changes in serum bicarbonate. However, Sargent's model was successful in accurately predicting bicarbonate concentrations during a more complex protocol with step-wise increase in dialysate bicarbonate [27]. All models except the original by Thews and Hutten [22] omitted the role of respiration and thus were unable to simulate differences in arterial and venous acid-base concentrations, especially those for $CO_2$. Further, all previous models have neglected the role of intracellular buffers in erythrocytes and muscle cells that have a significant effect on the whole system [28,29].

We propose a model of dissolved $CO_2$, bicarbonate, and oxygen ($O_2$) transport, describing comprehensively the effect of HD with bicarbonate-containing dialysis fluid on the acid-base chemistry of arterial and venous blood in different compartments, including tissue cells and interstitial fluid. The action of the lungs on the regulation of acid-base equilibria was described,

to allow simulating the effects of changes in the respiration parameters. The model was developed with the aim of being practical, using as inputs common clinical parameters that are possible to measure noninvasively in HD patients. This paper offers a description of the model and a demonstration of its feasibility in describing these processes, with simulations of common patterns observed in HD and specific clinical data. Finally, our results suggested a new hypothesis to explain the lack of equilibration between plasma and dialysis fluid bicarbonate.

## Methods

### Mathematical model

The model proposed is based on the $O_2$ and $CO_2$ storage and transport model developed by Andreassen and Rees [30], adapted to describe the changes in the acid-base chemistry of different body compartments induced by routine HD treatments using bicarbonate-containing dialysis fluid. All chemical reaction equilibrium constants and other physicochemical parameters were assumed as originally described [14,30] and are not reported here, whereas the patient-specific and treatment-specific parameters of the model were taken from the literature [26,31].

The basic structure of the model is the same as originally described [30]. The state variables are: the fraction of expired gas ($O_2$ and $CO_2$) in the lung compartment (lung capillaries and alveoli); the total $CO_2$ (bicarbonate plus dissolved $CO_2$) concentration, total $O_2$ concentration and the base excess ($BE$) in the arterial and mixed venous blood compartments; total $CO_2$ and total $O_2$ concentration in the tissue compartment (which unifies interstitial fluid and intracellular fluid). In total there are 4 interconnected compartments and 10 state variables defining the acid-base chemistry of each compartment (Fig 1).

We modified this model to include 3 additional state variables to describe the changes in the volume of arterial blood, venous blood, and interstitial fluid resulting from ultrafiltration of fluid during the HD treatment and interdialytic fluid gain.

Variables in each compartment were used to solve a system of equations with up to 19 unknowns, quantifying the equilibrium acid-base chemistry of blood and interstitial fluid. This model of blood chemistry was first developed by Rees and Andreassen [14]; the following is a brief outline of the quantitative relationships described in that model, while the equations are reported in Appendix A in S1 Text. Blood is assumed to consist of a fraction with red blood cells (erythrocyte volume) and without red blood cells (plasma volume). The total concentration of carbon dioxide in blood ($tCO_{2,b}$) is assumed to be the volume-weighted average of the total concentration of carbon dioxide in plasma and red blood cells. The total concentration of carbon dioxide in plasma is the sum of the concentrations of dissolved $CO_2$ and bicarbonate in plasma, and the total concentration of carbon dioxide in red blood cells is the sum of the concentrations of dissolved $CO_2$ and bicarbonate within red blood cells plus that of carbamated hemoglobin (hemoglobin with bound carbon dioxide moieties). The total concentration of oxygen in blood ($tO_{2,b}$) is similarly quantified. Non-bicarbonate buffers are present in plasma and interstitial fluid, representing large, negatively charged molecules, and phosphates, in protonated and non-protonated forms. Blood BE was calculated as the total buffer base minus the normal buffer base as defined elsewhere [14,32]. A similar approach was used to write a system of equations to calculate $tCO_2$ and $tO_2$ in the interstitial and tissue cells fluid compartments, although some more empirical relationships were used to describe the changes in interstitial and intracellular pH and base excess, due to the scarcity of information on the kinetics of these variables inside tissue cells [30] (Appendix B in S1 Text).

The solution of these systems is defined at each step of integration of the dynamic model (for each updated value of the state variables) to calculate the values of all the variables appearing in the state equations.

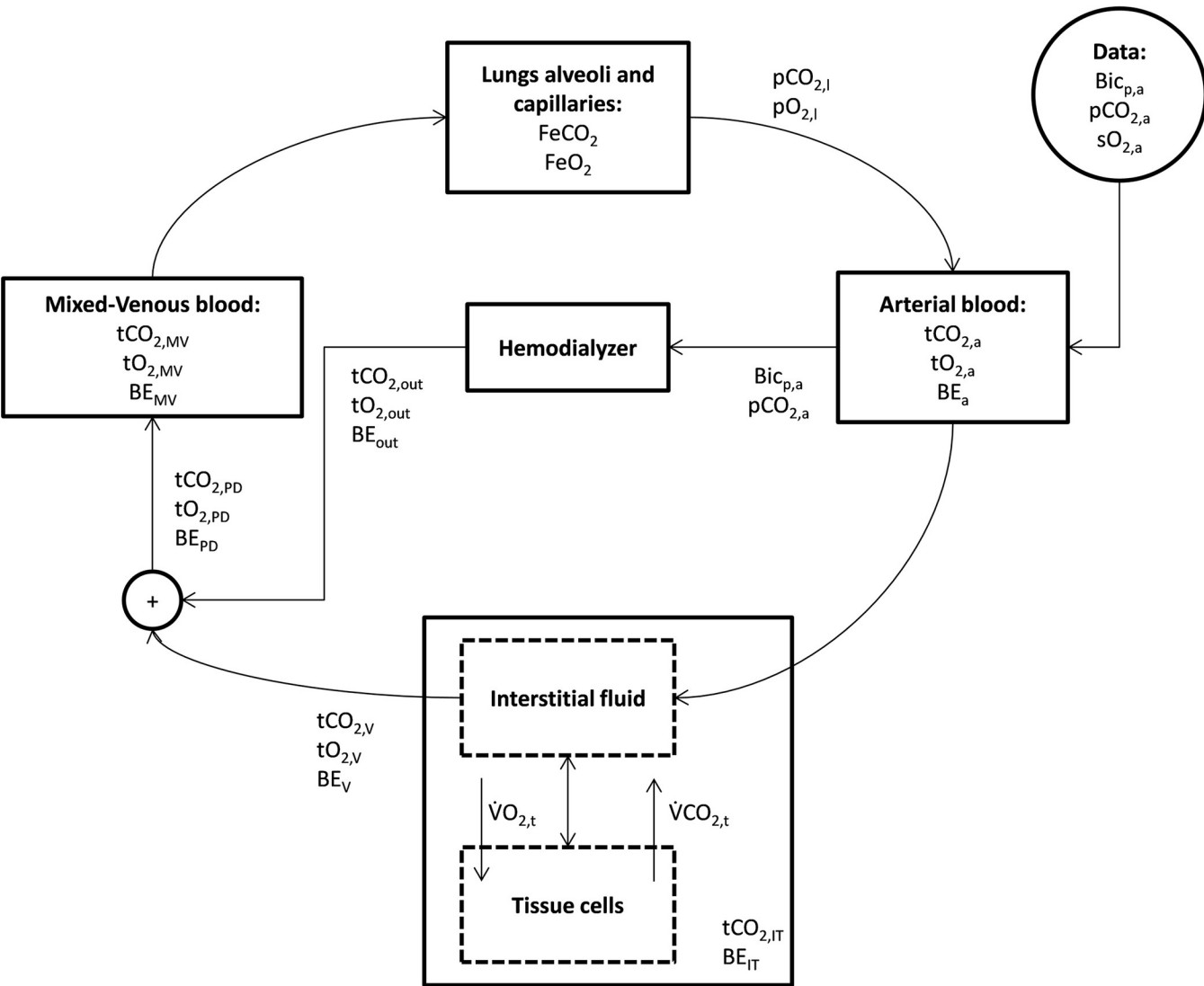

**Fig 1. General structure of the modified model of transport of carbon dioxide and oxygen in the human body, summarizing the input and outputs of the model compartments.** The state variables are reported for each of the four main compartments, and their value is calculated by solving the blood chemistry system of equations using the inputs for each compartment. See Appendix D in S1 Text for the explanation of the symbols.

The original whole-body model was developed mainly to quantify respiratory acid-base regulation, and all the variables related to blood chemistry depend on the input describing the respiration and metabolic steady states [30]. In order to apply the model to the HD patient data available, we introduced changes in the calculation of the steady-state of the model so that in pre-dialysis equilibrium conditions, all other variables could be calculated from the values of arterial partial pressure of $CO_2$ ($pCO_{2,a}$), arterial plasma bicarbonate concentration ($Bic_{p,a}$), and arterial oxygen saturation ($sO_{2,a}$). The latter is not regularly reported in published data [26,31] but, as in the original model [30], the knowledge of three variables was needed to define the chemistry of a compartment, so an initial value of $sO_{2,a}$ was assumed. Then, from the value of these three quantities in arterial blood, two other parameters were tuned, the tissue oxygen consumption ($\dot{V}O_{2,t}$) and the tissue $CO_2$ generation ($\dot{V}CO_{2,t}$), so that the initial solution of the time-dependent model would result in the observed pre-HD values of $pCO_{2,a}$, $Bic_{p,a}$

and $sO_{2,a}$. An explanation of these modifications to the steady-state calculation of the model is presented in Appendix C in S1 Text.

Most of the equations describing the time-dependency of the model are unchanged from the original proposed by Andreassen and Rees [30] and are summarized in Appendix D in S1 Text. The main differences consist in the introduction of the hemodialyzer, adding bicarbonate, dissolved $CO_2$ and acetate to blood, and the removal of the pulmonary shunt blood flow, considered negligible. We assumed an arterio-venous fistula access type. When blood passes through the dialyzer, the concentrations at the outlet of the filter for plasma bicarbonate ($Bic_{p, out}$) and partial pressure of $CO_2$ ($pCO_{2,out}$) are increased. $pCO_{2,out}$ is increased because of the presence of dissolved $CO_2$ in the dialysis fluid which diffuses to blood with a flow proportional to the gradient between plasma and dialysis fluid. $Bic_{p,out}$ increases because of two sources: the first is the diffusion flow of bicarbonate from the hemodialyzer; the second is the conversion of dialyzed acetate into bicarbonate.

$$Bic_{p,out} = Bic_{p,a} + \frac{D_{Bic}}{0.93 \cdot (1 - Hct) \cdot \dot{Q}_b} \left( \alpha_D \cdot Bic_d - Bic_{p,a} \right) + \frac{K_{Ac} \cdot Ac_p}{0.93 \cdot (1 - Hct) \cdot \dot{Q}_b} \quad (1)$$

$$pCO_{2,out} = pCO_{2,a} + \frac{D_{CO2}}{0.93 \cdot \dot{Q}_b} \left( pCO_{2,d} - pCO_{2,a} \right) \quad (2)$$

In the equations above, derived from the mass conservation principle, the subscript $d$ stands for dialysate, $p$ for plasma and $a$ for arterial. $D_{Bic}$ and $D_{CO2}$ are the dialysances for bicarbonate and dissolved $CO_2$, respectively. $K_{Ac}$ is a metabolic constant describing the conversion rate of acetate into bicarbonate, and it was assumed to be 0.65 L/min as determined in a previous study by Sargent et al [26]. $Ac_p$ is the plasma concentration of acetate, which is normally close to zero initially but quickly increases in patients during HD sessions because of diffusion from acetate-containing dialysis fluid. For simplicity, because previous studies showed that its value is relatively constant during HD and it reaches 99% of its plateau value in less than one hour of dialysis, we assumed $Ac_p$ to be constant, with values of 0.56 mmol/L and 0.325 mmol/L, when analyzing Sargent's data and Park's data, respectively [26,31]. $\dot{Q}_b$ is the flow rate in the blood channel of the extracorporeal circuit. It was assumed that the solute exchange at the two sides of the dialyzer membrane takes place between plasma and dialysis fluid, so for bicarbonate the term $0.93 \cdot (1 - Hct) \cdot \dot{Q}_b$ is the plasma water flow rate (0.93 is the correction for the water fraction of blood and $Hct$ is the haematocrit). Because dissolved $CO_2$ quickly diffuses across the erythrocyte's membrane, the full blood flow was used in Eq 2. The coefficient $\alpha_D$ corrects for the Donnan effect and is equal to 0.95 for bicarbonate and to 1 for $CO_2$. The value of $pCO_{2,d}$ was calculated with the Henderson–Hasselbalch equation assuming a pH equal to 7.2. Because of the low ultrafiltration rate compared to blood flow rate in these studies, for simplicity we assumed no convective transport of solute across the dialyzer membrane (however, it can be easily implemented by expanding Eqs 2 and 3).

Since clinical studies have shown that oxygen transport across the dialyzer is of minor magnitude [33], it was assumed that $tO_2$ is not significantly different at the two sides of the dialyzer ($tO_{2,out} = tO_{2,a}$).

With this assumption and Eqs 1 and 2, $Bic_p$, $CO_{2,p}$ and $tO_2$ are known for the blood at the dialyzer outlet, and it is possible to solve the system of equations (Appendix A in S1 Text) to calculate the total $CO_2$ concentration ($tCO_{2,out}$), which is then mixed with the venous blood

(subscript $V$) that bypassed the fistula, to give the post-dialyzer ($pd$) blood:

$$tCO_{2,pd} = tCO_{2,v} \cdot \left(1 - \frac{\dot{Q}_b}{\dot{Q}_{CO}}\right) + tCO_{2,out} \cdot \frac{\dot{Q}_b}{\dot{Q}_{CO}} \qquad (3)$$

Similar equations are written for $tO_{2,PD}$ and $BE_{PD}$; the three post-dialyzer values are used as the inlet of the mixed-venous ($mv$) compartment in the corresponding state-equations:

$$\frac{dtCO_{2,mv}}{dt} = \frac{\dot{Q}_{CO}}{V_{mv}} \left(tCO_{2,PD} - tCO_{2,mv}\right)$$

$$\frac{dtO_{2,mv}}{dt} = \frac{\dot{Q}_{CO}}{V_{mv}} \left(tO_{2,PD} - tO_{2,mv}\right) \qquad (4)$$

$$\frac{dBE_{mv}}{dt} = \frac{\dot{Q}_{CO}}{V_{mv}} \left(BE_{PD} - BE_{mv}\right)$$

where $V_{mv}$ is the volume of the mixed-venous blood compartment and $\dot{Q}_{CO}$ is the cardiac output. The state equations for the other compartments are written with a similar approach and are reported in Appendix D in S1 Text.

Volumes of interstitial and blood compartments were assumed to change linearly throughout the session; other volumes were maintained constant, consistent with the removal of fluid from only extracellular sources. In blood only plasma volume decreases, whereas the volume of erythrocytes is constant. A constant fraction of total ultrafiltration volume (80%) was assumed to come from the interstitial fluid, and the remaining 20% to come from mixed venous and arterial pools, proportional to their volume ratio before the treatment. The mass of total non-bicarbonate buffer base in plasma and interstitial fluid was kept constant during the session, but its concentration increased according to the reduction in the respective volumes. The volume of the tissue cells compartment was assumed to be constant.

The three volume state equations added to the model have thus the form:

$$\frac{dV_x}{dt} = -\rho \cdot \dot{Q}_{uf} \qquad (5)$$

where $V_x$ can be arterial, mixed venous or interstitial volume, $\rho$ is the fraction of total ultrafiltration calculated as described in the previous paragraph, and $\dot{Q}_{uf}$ is the ultrafiltration rate calculated as the ratio between ultrafiltration volume and session length. The volumes of blood and interstitial fluid were assumed to increase linearly during interdialytic intervals.

## Model simulations and clinical data

The model was implemented in MatLab (MathWorks, Natick, MA, USA). The ability of the model to predict published clinical data was assessed by fitting the model to that reported in the studies by Sargent [26] and Park (study 1a) [31]. Patient-specific and treatment-specific parameters were taken from the published data whenever possible, and the quantities not reported in the studies were either assumed equal to what used in the original formulation of the model or fixed based on common values for the dialysis population (Table 1).

The fitting was carried out by tuning two model parameters to minimize the sum of residuals ($SR$) between the data and the model's outputs. The total $SR$ was calculated summing the

**Table 1. Patient-specific and treatment-specific parameters for the two clinical studies [26,31] to which the model was applied.** The values used here are the average of the data reported. For patient-specific parameters, initial values are reported unless otherwise specified. Measurements missing in the datasets were taken from the original description of the model or assumed based on standard values for the patient population.

| Parameter | Symbol | Sargent et al | Park et al |
|---|---|---|---|
| Plasma bicarbonate (mmol/L) | $Bic_p$ | 21.3 | 23.07[a] |
| Plasma $CO_2$ partial pressure (mmHg) | $pCO_{2,p}$ | 36.00 | 38.85[a] |
| Blood hemoglobin (mmol/L) | $Hb$ | 7.2 | 7.07 |
| Erythrocyte hemoglobin (mmol/L) | $Hb_e$ | 20.67[b] | 21.4[b] |
| Oxygen saturation | $sO_2$ | 0.97[c] | 0.97[c] |
| Cardiac output (L/min)* | $\dot{Q}_{CO}$ | 5.0[d] | 5.0[d] |
| Net acid production rate (mEq/Kg BW/day)* | $H_t$ | 1.0[d] | 1.0[d] |
| Plasma non-bicarbonate buffer concentration (mEq/L) | $tNBB_{p,0}$ | 23.5[d] | 23.5[d] |
| Intracellular total non-bicarbonate buffer capacity (mmol/pH/kg $H_2O$) | $\beta_{tNBBt}$ | 30[d] | 30[d] |
| Body weight (Kg) | $BW_0$ | 71 | 85 |
| Total body water volume (L) | $TBW_0$ | 33.3 | - |
| Extracellular volume (L) | $ECF_0$ | 12.3 | - |
| Interstitial volume (L) | $V_{i,0}$ | 8.8[b] | 14[c] |
| Tissue intracellular volume (L)* | $V_t$ | 21[b] | 14[c] |
| Hematocrit | $Hct_0$ | 0.348 | 0.33[c] |
| Blood volume (L) | $V_{b,0}$ | 5.29 | 5.0[c] |
| Plasma volume (L) | $V_{p,0}$ | 3.45[b] | 3.35[b] |
| Arterial/venous volume ratio | | 0.149[d] | 0.149[d] |
| HD duration (min)* | $t_{HD}$ | 209 | 224 |
| Ultrafiltration volume (L)* | $UFV$ | 1.8 | 2.8 |
| Dialysate bicarbonate (mmol/L)* | $Bic_d$ | 32 | 37 |
| Dialysate $CO_2$ partial pressure (mmHg)* | $pCO_{2,d}$ | 82.90[e] | 95.85[e] |
| Dialyzer blood flow (mL/min)* | $\dot{Q}_b$ | 400 | 385 |

[a] value obtained from graphs using WebPlotDigitizer (Ankit Rohatgi, https://automeris.io/WebPlotDigitizer)

[b] calculated

[c] assumed

[d] taken from references [14,30]

[e] calculated with the Henderson-Hasselbalch equation assuming dialysate pH = 7.2 [33]

* constant parameter.

residuals for arterial plasma bicarbonate concentration and partial pressure of $CO_2$:

$$SR = \sum_i \left( \frac{Bic_{\text{data,i}} - Bic_{\text{model,i}}}{Bic_{\text{data,i}}} \right) + \sum_i \left( \frac{pCO_{2,\text{data,i}} - pCO_{2,\text{model,i}}}{pCO_{2,\text{data,i}}} \right) \tag{6}$$

The subscript $i$ indicates the i-th measurement collected during the HD session. Measurements of plasma bicarbonate and $pCO_2$ were taken in the Sargent study at t = 0, 15, 30, 60, 90, 120, 209 minutes and in the Park study at $t$ = 0, 15, 45, 90, 135, 180, 224 minutes. The parameters fitted were the dialysances for bicarbonate ($D_{Bic}$) and dissolved $CO_2$ ($D_{CO2}$). The estimation of the parameters was carried out with a nonlinear least squares method (function *lsqnonlin* in MatLab, used also for the estimation of steady-state parameters).

Additional simulations included the simulation of a weekly HD cycle, comprised of three sessions of similar duration and post-dialytic intervals of 48, 48 and 72 hours, respectively,

carried out to test the feasibility of simulating interdialytic changes in acid-base chemistry. Parameter values taken from the Sargent dataset were used in this case.

### Sensitivity analysis

Local sensitivity analysis of the model was carried out for the dialysances and for the parameters reported in Table 1, to assess their impact on selected outputs of the model. The sensitivity indices were calculated using the one-at-a-time (*OAT*) method. More details on the methods and the results of this analysis is reported in the Appendix E in S1 Text.

## Results

### Simulation of clinical data

Table 2 shows the parameters estimated by the model to fit the model predictions to the clinical data from the studies of Sargent et al [26] and Park et al [31]. Tissue oxygen consumption ($\dot{V}O_{2,t}$) and CO$_2$ generation ($\dot{V}CO_{2,t}$) were tuned to fit the initial state of the model to the values of plasma bicarbonate and pCO$_2$ (Table 1), while the dialysances $D_{Bic}$ and $D_{CO2}$ were tuned to fit the simulated bicarbonate and pCO$_2$ profiles during the session. The values of $\dot{V}O_{2,t}$ and $\dot{V}CO_{2,t}$ were considered constant during the simulated HD sessions.

The output of the model's fit to Sargent's data is shown in Fig 2, whereas that for Park's data are shown in Fig 3. While the arterial plasma variables of the model were fitted to the data, the model predicts concentration profiles of mixed venous plasma as well as those in other compartments (erythrocytes, interstitial fluid, tissue cells); these additional results shown in the figures are examples of the variety of the predictions obtainable from the model. Several trends are consistent from the model predictions using both data sets. For example, mixed venous plasma levels of bicarbonate concentration and pCO$_2$ are higher than those in arterial plasma, and both bicarbonate concentration and pCO$_2$ are substantially lower in erythrocytes than in plasma. Further, model-predicted acid-base parameters in the interstitium are similar to those in mixed venous blood. These relationships are like those predicted by this same model under normal physiological conditions [30]. Integration of Eqs 1 and 2 allowed us to calculate the amount of base and dissolved CO$_2$ transferred from dialysis fluid to the patient according to the model. For Sargent's data they are equal to 181.5 mmol (of which 76.1 are from acetate) and 25.1 mmol, respectively, and for Park's data they are equal to 179.8 mmol (of which 47.3 are from acetate) and 19.8 mmol, respectively. Despite the different combinations of fixed parameters tested, the model with constant O$_2$ consumption, CO$_2$ generation and

**Table 2. Parameters estimated by the model.** The values estimated to fit the data from the studies by Sargent [26] and Park [31] are compared with the values assumed in the original model description by Andreassen and Rees [30] (only for the steady-state parameters $\dot{V}O_{2,t}$ and $\dot{V}CO_{2,t}$). The dialysances $D_{Bic}$ and $D_{CO2}$ are shown first for the simulations with constant respiration rate and then when a linear increase in minute ventilation ($\Delta\dot{V}_E$) was assumed.

| Parameters | Sargent et al | Park et al | Andreassen and Rees | $\Delta\dot{V}E$ |
|---|---|---|---|---|
| $\dot{V}O_{2,t}$ *(mL/min)* | 283.6 | 280.1 | 253 | |
| $\dot{V}CO_{2,t}$ *(mL/min)* | 171.3 | 179.8 | 222 | |
| $D_{Bic}$ *(mL/min)* | 124.7 | 78.8 | - | 0 |
| $D_{CO2}$ *(mL/min)* | 87.0 | 50.9 | - | 0 |
| $D_{Bic}$ *(mL/min)* | 164.4 | 142.8 | - | 14.0% |
| $D_{CO2}$ *(mL/min)* | 131.9 | 139.7 | - | 22.8% |

$\dot{V}O_{2,t}$–tissue oxygen consumption rate; $\dot{V}CO_{2,t}$–tissue CO$_2$ generation rate.

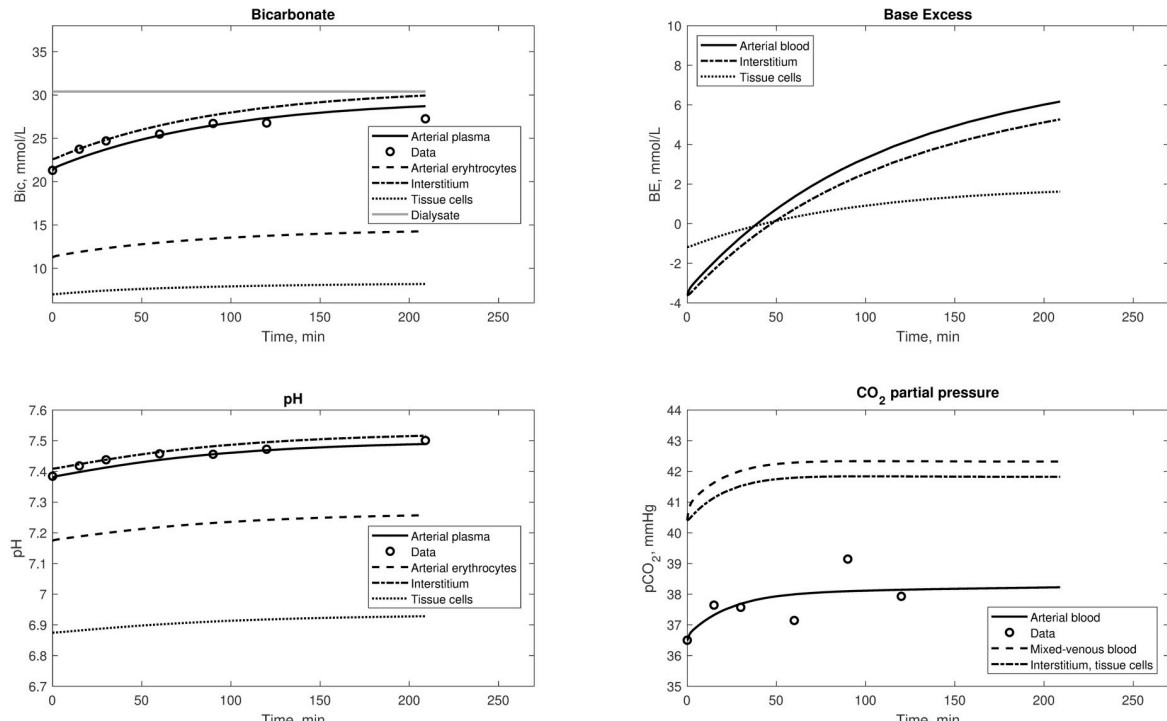

**Fig 2. Simulations of several variables of the model, fitted to the data from the study by Sargent et al [26].** The dialysate concentration of bicarbonate was corrected for the Gibs-Donnan effect.

respiration rates were not able to match the pattern of $pCO_2$ data, although the small variability of the values in the data assured a small overall error.

## Changes to the respiratory regulation of $pCO_2$

For each initial steady-state defined by the model's fixed parameters and $\dot{V}O_{2,t}$ and $\dot{V}CO_{2,t}$, the minute ventilation of the patient was calculated ($\dot{V}_E$, the volume of air breathed per minute). This parameter was assumed constant during the HD session, for lack of data, but an increase in respiration rate during dialysis has been observed in clinical studies [34–36]. Additional model simulations were therefore carried out with the assumption of a linear increase in $\dot{V}_E$, and the dialysances $D_{Bic}$ and $D_{CO2}$ were newly fitted to the data, together with the optimal increase in $\dot{V}_E$, expressed as a percentage of the initial value ($\Delta \dot{V}_E$). These results are presented in Fig 4. The baseline values of $\dot{V}_E$ were 3.55 and 3.49 L/min for Sargent's and Park's data, respectively. The new estimated parameters are shown in Table 2. The assumption of linearly increasing $\dot{V}_E$ allowed the model to achieve a better fit of $pCO_2$ data. The increase in the estimated dialysances also brought an increase in the added bicarbonate and $CO_2$ to the patient. For Sargent's data the new values were 219.2 mmol (of which still 76.1 mmol were from acetate) and 38.4 mmol, respectively, and for Park's data they were equal to 279.2 mmol (47.3 mmol were from acetate) and 54.0 mmol, respectively.

## Theoretical simulation experiments

Fig 5 depicts a cycle of consecutive HD sessions, following a standard 3-2-2 schedule (3 days pre-dialytic interval before the first session of the week, 2 days before the following sessions).

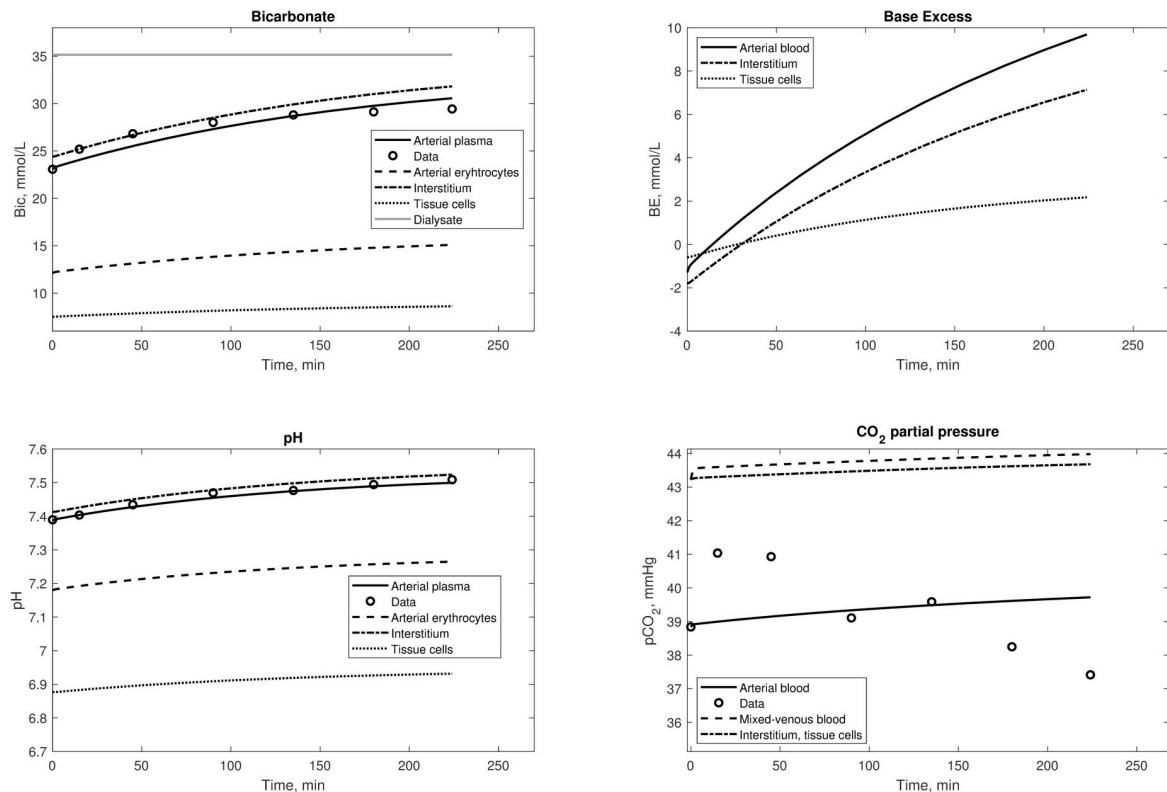

**Fig 3. Simulations of several variables of the model, fitted to the data from the study by Park et al [31].** The dialysate concentration of bicarbonate was corrected for the Gibs-Donnan effect.

The model was able to correctly describe the alkalinization of the patient during each session, and consequent acidification in the intradialytic periods. Values of $pCO_2$ were maintained constant by normal lung function except during HD treatments with the influx of bicarbonate and dissolved $CO_2$ from the dialysis fluid. A cyclical steady-state was obtained, with the value of the output at the end of the cycle equal to the initial value, by adjusting the net acid generation rate ($H_t$) to be in equilibrium with the infusion of buffer base during each session. For this simulation, $H_t$ was tuned to 0.043 mL/min.

## Discussion

This work describes a mathematical model simulating the time-dependent effect of HD on acid-base chemistry within the patient's body. The current model is based on existing acid-base chemistry and whole-body compartmental models proposed in twin papers by Rees and Andreassen [14,30]. We adapted the model to describe the changes in acid-base chemistry of the body compartment during a HD session, modifying its framework to allow the use of clinical data of HD patients as inputs, and adding equations to describe the transport of bicarbonate and dissolved $CO_2$ and removal of fluid by the hemodialyzer. With such modifications, the model was able to describe the effects of hemodialysis on a number of acid-base related quantities (pH, bicarbonate and hemoglobin concentrations, partial pressures of $CO_2$ and $O_2$) simultaneously, in both blood and extravascular compartments.

The model developed by Rees and Andreassen was chosen as the base for this work because of its relative simplicity in the face of its comprehensive approach to the description of the acid-base status in the body. We acknowledge that more complete models describing

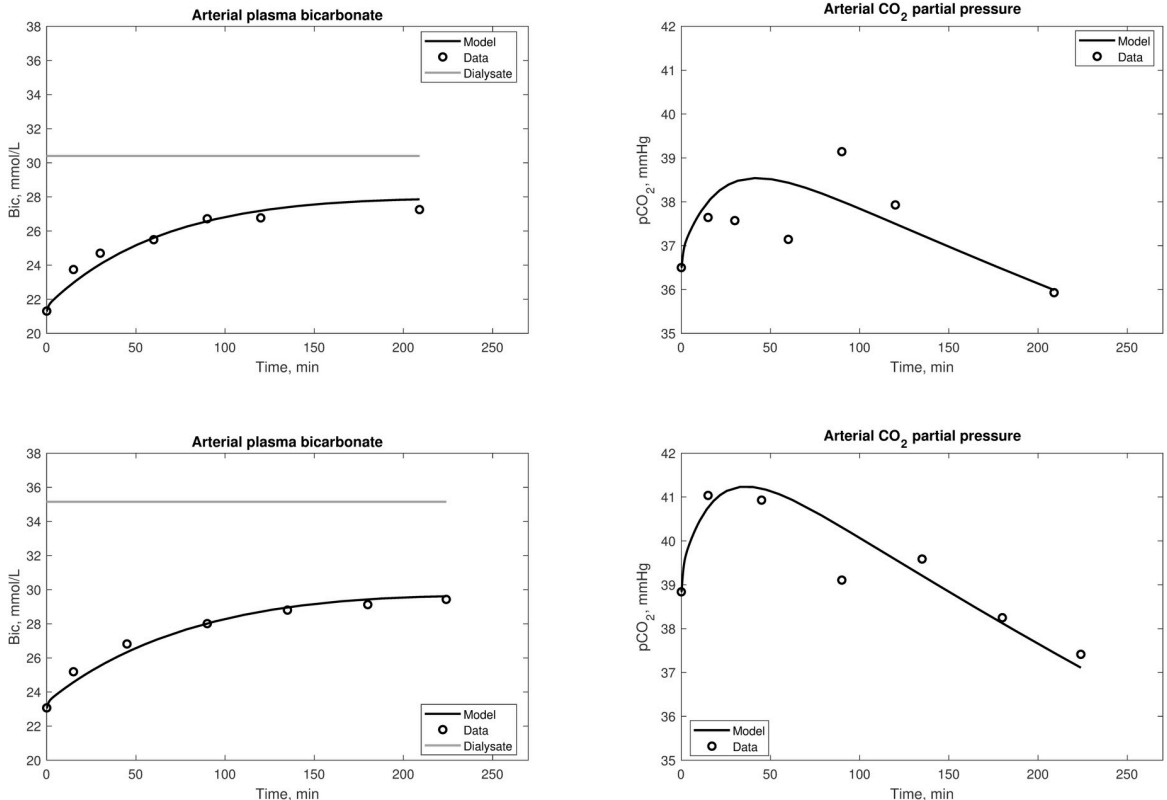

**Fig 4. Simulations with a linearly increasing minute ventilation during the HD session.** The increase was up to 11% more than the initial value for Sargent data (top panels) and 22% for Park data (bottom panels). The dialysate concentration of bicarbonate was corrected for the Gibs-Donnan effect.

physiological acid-base phenomena exist, but their application to the simulation of HD can be too complex for practical use [21,37,38]. Conversely, existing models applied specifically to the study of HD tend to neglect important mechanisms such as intracellular and non-bicarbonate buffers and respiratory regulation [19,26]. Most models neglect the transport of dissolved $CO_2$ from dialysis fluid, whose importance we discussed in a recent paper [39], and whose tangible impact has been revealed by symptoms such as HD-driven intra-dialytic acidosis [18,40]. Furthermore, the presence of intracellular buffers, and transport of mobile buffers such as bicarbonate between intracellular and extracellular fluids directly affects the acid-base equilibrium and should be taken into account in a mathematical description of these phenomena [41]. It was thus of importance that our modeling effort would include a physiology-based description of the interactions between intracellular and extracellular compartments, accounting for the chemical reactions involving bicarbonate and other non-bicarbonate buffers.

For this theoretical study we applied the mathematical model to the simulation of clinical profiles of plasma bicarbonate, pH, and $pCO_2$ reported in the published works by Sargent et al [42] and Park et al [31]. The model was fitted to the data in two steps. First the initial value of the state-variables was identified by fitting two parameters ($\dot{V}O_{2,t}$ and $\dot{V}CO_{2,t}$) to assure that the stead-state solution of the model would reflect the pre-dialysis bicarbonate and $pCO_2$ data; successively, the profiles during HD were fitted by tuning the dialysances for bicarbonate ($D_{Bic}$) and dissolved $CO_2$ ($D_{CO2}$). The estimated values of $\dot{V}O_{2,t}$ and $\dot{V}CO_{2,t}$ were similar to those assumed by Andreassen and Rees [30]. $D_{Bic}$ and $D_{CO2}$ were selected for tuning after the

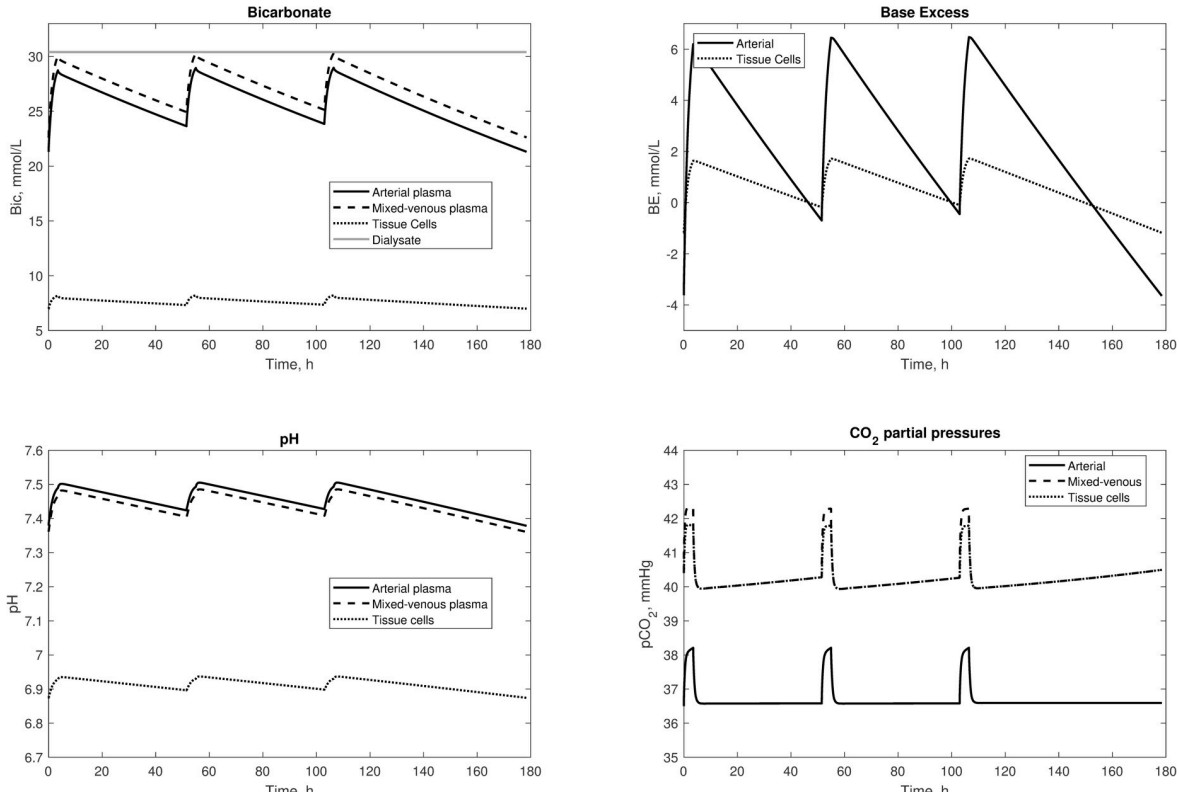

**Fig 5. Simulations of a week-long cycle of HD sessions.** Two postdialytic intervals were of 48 hours and one was of 72 hours, during which fluid volumes accumulates and bicarbonate buffer is depleted due to an endogenous acid production rate of 0.039 mmol/day per Kg of body mass. The dialysate concentration of bicarbonate was corrected for the Gibs-Donnan effect.

sensitivity analysis (Appendix E in S1 Text) showed that they were among the main determinants of arterial plasma bicarbonate concentration and $pCO_2$ (pH was not considered, as its value depends on bicarbonate and $pCO_2$ via the Henderson-Hasselbalch equation).

Previous models of acid-base chemistry during HD have been limited to describing intradialytic changes in acid-base parameters [19,22,24,26]. The current model is the first to describe the behavior of the acid-base parameters both during and between HD treatments (Fig 5). As expected, the blood is loaded with alkali during the HD session, while the metabolic acids generated during interdialytic intervals acidify the system via protein catabolism, thereby decreasing pH and bicarbonate content [43]. The simulations of the weekly HD cycle demonstrated that a cyclic steady-state between weeks of treatment can be achieved by tuning the endogenous net acid production rate of the patient so that the output values after the third post-dialytic interval would match the first pre-dialytic ones. This suggests that it may be possible to use the model described here to estimate the acid generation rate from such observed clinical data. Conversely, by knowing the acid generation rate during a cycle, the dose of bicarbonate dialysis necessary to achieve the desired post-dialysis blood chemistry status at the end of the week can be more accurately estimated. It must be noted that, for simplicity, the model neglects the effects of residual renal clearance or additional buffer stores (such as in bone); in simulations with a long timescale this might introduce significant errors, although it is likely that these errors can be ignored for intradialytic simulations.

After tuning the selected parameters, our model was able to reproduce with good accuracy the arterial plasma bicarbonate concentration-time profiles present in the two datasets

**Table 3. Dialysances calculated from measured inlet and outlet dialyzer concentrations in published studies.**

|  | $D_{Bic}$ (D) | $D_{Bic}$ (B) | $D_{CO2}$ (D) | $D_{CO2}$ (B) | $Q_b$ | $Q_d$ |
|---|---|---|---|---|---|---|
| *Sombolos et al [33]* | 133.1 | 223.1 | 447.0 | 177.1 | 300 | 700 |
| *Symreng et al [36]* | 241.4 | 220.7 | 297.8 | 188.4 | 400 | 500 |

$D_{Bic}$–bicarbonate dialysance; $D_{CO2}$ –dissolved $CO_2$ dialysance; D–calculated from dialysate side; B–calculated from blood side. $Q_b$–blood flow rate; $Q_d$–dialysate flow rate. Values in mL/min.

analyzed. The interpretation of our estimated values of the dialysances of bicarbonate and $CO_2$ is, however, complicated by the interconversion of the two species, as already emphasized by several authors [33,39,44]. Several examples of this problem can be found in the literature. In a recent editorial in the American Journal of Physiology [45] Gennari and Sargent discussed the data described by Park et al [31], and observed that calculating the dialysance of bicarbonate from the concentration measurements yielded a value $D_{Bic}$ = 205 mL/min. However, they suggested that a lower $D_{Bic}$ would have been necessary, 104 mL/min, to obtain the net alkali transport from dialysis fluid to blood reported by Park. Only few studies reported concentrations at the inlet and outlet of a dialyzer for both bicarbonate and $pCO_2$, allowing to calculate dialysances using the classic formula. Sombolos et al made such measurements 5 minutes into the HD session [33], while Symreng et al took samples at 60 minutes [36]; the dialysances calculated from their data are shown in Table 3. Some anomalies are immediately apparent, such as $D_{CO2}$ reported by Sombolos being higher than the HD circuit blood flow rate. However, the values described by Sombolos were calculated using dialysate measurements: repeating the same calculation from blood values yields significantly lower values for $CO_2$ (Table 3). Although the blood-side dialysances reported for $CO_2$ in Table 3 are closer to what was estimated by our model, these observations suggest that the classic method of dialysance calculation is ill-suited for bicarbonate and $CO_2$ (bicarbonate dialysances also would be inflated by the additional bicarbonate from acetate metabolism). By contrast, the dialysances appearing in Eqs 1 and 2 of our model are direct expression of the rate of diffusive transport across the dialyzer, and thus should probably not be directly compared.

That respiration rate increases during the HD session is open to debate. An increase in respiration rate to ventilate the excess $CO_2$ is actually to be expected [43]; however, only a small increase was reported in clinical studies during a HD session, and it was often not statistically significant [34,35]. Symreng measured an average increase in minute ventilation of 12.3% in 5 patient undergoing HD, comparable to the increase estimated by our model. The fitting of the simulated profiles of $pCO_2$ to the data was significantly improved when the minute ventilation $\dot{V}_E$ was allowed to increase during the session (Fig 4). Because of the lack of information on the changes in respiration rate during the clinical sessions, a linear increase in minute ventilation was assumed. We found that an increase corresponding to 1.3 breaths/minute in Sargent's data and 1.7 breaths/minute in Park's data was enough to achieve a close fit of the data, and reproduce the characteristic small increase in $pCO_2$ during the first hour of HD observed in clinical studies [26,46–48].

The newly estimated dialysances were higher than in the baseline model for both data sets since an increase in transport was necessary to balance the higher exhalation of $CO_2$ from the lungs during the HD session. This significantly affected the mass balance for both bicarbonate and $CO_2$ (but not for acetate, whose contribution remained constant with the constant plasma concentration assumed in the model). The net alkali mass transported across the dialyzer reported by Park was equal to 155 mmol, while Sargent reported 203 mmol [26,31]; using Sargent's data, our model predicted values ranging from 181 to 219 mmol, depending if the

respiration rate was assumed constant or increasing during the HD session. For Park's data, predicted net alkali balance changed from being close to the published value when respiration rate was constant, to being almost double when a 23% $\Delta \dot{V}_E$ was estimated (279 mmol). However, it is possible that the model might be overestimating the increase in $\dot{V}_E$; assuming only 13% for Park's data resulted in significantly lower transport parameters and mass balance ($D_{Bic}$ = 116 mL/min, $D_{CO2}$ = 100 mL/min, net alkali transport = 237 mmol) with an increase in the relative prediction error for pCO₂ of only 0.3%. Therefore, it is possible that with a more accurate function describing changes in the respiration during HD, a more accurate mass balance could be predicted by the model.

These simulations show anyway how sensitive the whole system is to changes in the respiration and demonstrate the possibility of using the model to simulate their effects on the regulation of the acid-base equilibria.

In clinical practice a lack of equilibration between plasma and dialysate bicarbonate concentration is often observed, despite a leveling-off of plasma bicarbonate [15–17]; Gennari et al [49] have recently summarized some possible explanations for this lack of equilibrium. In one case [26], Sargent et al proposed that this lack of equilibration was the result of increased endogenous organic acid production; however, recent data from Park et al [31] do not support endogenous acid production as the dominant mechanism. Another study suggested that a release of H+ ions was caused by a delayed activation of bone buffering, but without direct evidence to verify this hypothesis [50]. Recently, Wolf [19] suggested that the leveling off of bicarbonate could be explained as a consequence of the electroneutrality principle. We propose yet an alternative possible explanation based on the current described model. According to our model, the main cause of the lack of equilibration of blood and dialysate bicarbonate levels appears to be dissolved $CO_2$ transport from dialysis fluid to blood. The acidifying effect of dialysate-wrought dissolved $CO_2$ has been already addressed in the past [18,40]; our results suggest that progressive fall in pH resulting from the increasing pCO₂ somewhat counteracts the increase in bicarbonate concentration. If, for example, the dialysance $D_{CO2}$ is set to zero in our model, that is eliminating dissolved $CO_2$ transport from the dialysis fluid, equilibration is completely achieved, except for a small offset caused by the net endogenous acid generation rate (Fig 6, when setting acetate to zero). During the HD session this results in a slower increase in bicarbonate and other bases (non-bicarbonate buffer and hemoglobin in non-protonated forms), until reaching an equilibrium in which the blood concentration of these bases levels off. This equilibrium is preserved by the stability of the flow of dissolved $CO_2$ from dialysis fluid (Fig 6), in turn maintained by the exhalation of $CO_2$ from the lungs (Equations A39 and A40, Appendix D in S1 Text) which dampens the increase in arterial $pCO_2$. The flattening of bicarbonate profiles, incomplete in simulations with constant respiration rate (Figs 2 and 3), was achieved when an increase was assumed in $\dot{V}_E$ (Fig 4). Simulations with different dissolved $CO_2$ concentrations in the dialysis fluid showed that the plasma bicarbonate level reached at the end of HD is inversely proportional to the former, all other factors being equal. It must be noted that, because of the formulation of the original model of Rees and Andreassen, changes in quantities such as bicarbonate, pH (H⁺ concentration), and dissolved $CO_2$ in plasma and erythrocytes are not calculated as result of modelling of their transport flows, but rather as solutions of a series of steady-states for the chemistry equations of each blood and fluid compartment. As such, it is difficult to directly assess from our modeling results the origin of the H⁺ anions titrating the dialyzed bicarbonate.

Our study has several additional limitations. Many subject-specific parameters were missing in the data to which the model was applied and had to be chosen based on common sense or known literature. Plasma/erythrocyte/hemoglobin biochemistry was assumed the same as

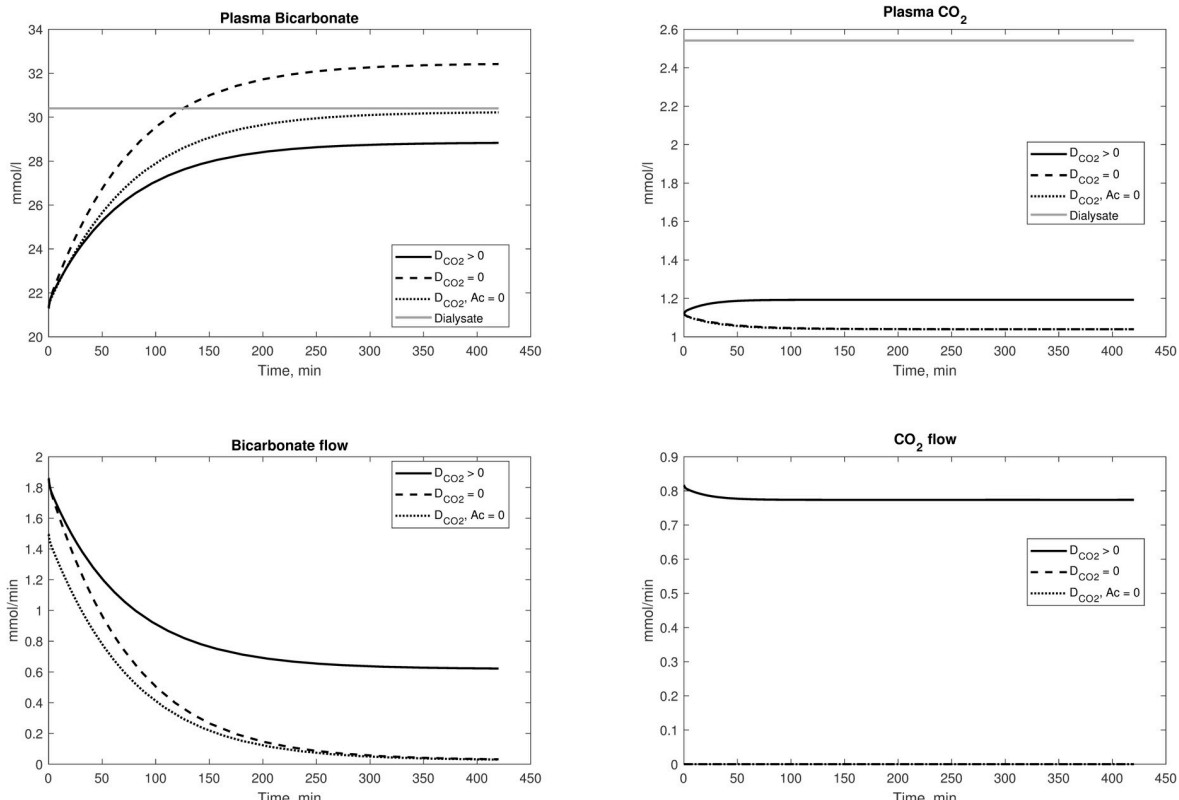

**Fig 6. Effects of decreasing $D_{CO2}$.** (Top) Bicarbonate and dissolved $CO_2$ concentrations in Sargent data (assuming constant respiration rate), with dialysate concentration of bicarbonate corrected for the Gibs-Donnan effect. The simulated session was extended to 420 minutes to better show the leveling off of bicarbonate concentration profile, and the effects of hemoconcentration were removed by setting the value of UF to zero. Note that if acetate concentration is not set to zero (dashed line), plasma bicarbonate can exceed dialysate concentration because of the constant inflow from acetate, which in the model is independent of other variables. (Bottom) Solute flow rates from dialysis fluid to plasma.

normal subjects, as described by Rees and Andreassen [14], therefore the model ignores possible changes caused by uremia. Furthermore, we assumed that only bicarbonate and dissolved $CO_2$ concentrations are affected by passage through the dialyzer; however, this can be a reasonable simplification because of the much higher relative changes observed in bicarbonate and $pCO_2$ compared to e.g. oxygen partial pressure [33]. Time-dependent transport of acetate from dialysis fluid to blood was simplified, opting instead to describe acetate plasma levels as constant. While this is clearly an oversimplification, we deemed it reasonable in the light of clinical and computational evidence [26,31]; anyway, simulations performed using a time-dependent acetate concentration as described in [26] resulted in only small changes of the estimated parameters (results not shown). Finally, our model approaches the study of acid-base homeostasis with the "classical" view of bicarbonate and base excess, contrary to the approach spearheaded by Stewart based on the electroneutrality principle [51], and employed by other investigators [19,52]; thus, we do not consider the transport of other ions, such as sodium, potassium and chloride, that are also altered by HD. This simplification was necessary to keep the complexity of the model low and avoid the need of more assumptions on the kinetics of other ions without such data; however, the two approaches can be seen as complementary [53].

To conclude, the model of acid-base chemistry and transport proposed in this study shows promise to describe the alterations in acid-base balance within the patient during HD and

managed to provide a close fit of clinical data without overparametrization, despite the lack of information on many patient-specific parameters. Our results suggest that the importance of the lungs on the preservation of the acid-base homeostasis during HD cannot be neglected even when describing metabolic acidosis. Given the sensitivity of the whole acid-base buffer system to changes in $pCO_2$, any further modeling efforts should strive to describe the respiratory regulation of $CO_2$ levels in the patient. Even a relatively ignored phenomenon, such as the transport of dissolved $CO_2$ from dialysis fluid to blood, might unexpectedly be one of the main determinants of the kinetics of bicarbonate during HD and of the post-dialysis bicarbonate plasma concentration. Overall, the strong physiological and physicochemical bases of the model, and the wide array of outputs available, make it useful for a more complete description of acid-base disturbances during dialysis.

## Supporting information

**S1 Text. Appendices.** Equations of the model (appendices A, B, C, D) and results of the sensitivity analysis (appendix E).
(PDF)

## Author Contributions

**Conceptualization:** Jacek Waniewski, John K. Leypoldt.

**Formal analysis:** Mauro Pietribiasi.

**Funding acquisition:** John K. Leypoldt.

**Investigation:** Mauro Pietribiasi, John K. Leypoldt.

**Methodology:** Mauro Pietribiasi, John K. Leypoldt.

**Software:** Mauro Pietribiasi.

**Supervision:** Jacek Waniewski, John K. Leypoldt.

**Validation:** Mauro Pietribiasi.

**Visualization:** Mauro Pietribiasi.

**Writing – original draft:** Mauro Pietribiasi.

**Writing – review & editing:** Mauro Pietribiasi, Jacek Waniewski, John K. Leypoldt.

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
