## [Decision Letter · Decision Letter 0]

10 Oct 2022

PONE-D-22-15261Mathematical modelling of bicarbonate supplementation and acid-base chemistry in kidney failure patients on hemodialysisPLOS ONE

Dear Dr. Pietribiasi 

Thank you for submitting your manuscript to PLOS ONE. After careful consideration, we feel that it has merit but does not fully meet PLOS ONE’s publication criteria as it currently stands. Therefore, we invite you to submit a revised version of the manuscript that addresses the points raised during the review process.

We look forward to receiving your revised manuscript.

Kind regards,

Emre Avci

Academic Editor

PLOS ONE

“This work was supported by the Polish National Science Center (grant number 2017/27/B/ST7/03029).”

“This work was funded by the Polish National Science Center (grant number: 2017/27/B/ST7/03029). JKL was the beneficiary of the grant. MP was employed in the grant. The funders had no role in study design, data collection and analysis, decision to publish, or preparation of the manuscript.”

Reviewers' comments:

Reviewer's Responses to Questions

**Comments to the Author**

1. Is the manuscript technically sound, and do the data support the conclusions?

Reviewer #1: Partly

Reviewer #2: Yes

2. Has the statistical analysis been performed appropriately and rigorously? 

Reviewer #1: N/A

Reviewer #2: Yes

3. Have the authors made all data underlying the findings in their manuscript fully available?

Reviewer #1: Yes

Reviewer #2: Yes

4. Is the manuscript presented in an intelligible fashion and written in standard English?

Reviewer #1: Yes

Reviewer #2: Yes

5. Review Comments to the Author

Reviewer #1: The authors have adapted a 4-compartment (plasma, erythrocytes, interstitial fluid and intracellular fluid) model of acid-base regulation in individuals with normal kidney function to analyze the acid base response to rapid alkalinization during hemodialysis. In contradistinction to earlier models, they also include the effect of respiratory control over the partial pressures of CO2 and oxygen. They have evaluated their model using data obtained in 2 patient studies in which blood bicarbonate concentration, pH and PCO2 were measured at timed intervals during treatment. They conclude that their model fits the clinical measurements well and assert that they have uncovered fresh insights into the complex acid-base interactions during hemodialysis. My comments are below.

1. A significant flaw in their model is to ignore the role of acetate influx and metabolism in bicarbonate addition to the patient during hemodialysis. They justify their lack of inclusion of by citing evidence that variations in bath acetate have no effect on the end-dialysis acid-base status in patients receiving hemodialysis. While this is true, it doesn’t mean that acetate influx and metabolism plays no role in contributing bicarbonate to the ECF fluid. Sargent et al have shown that 1/3 of the net bicarbonate added is from acetate influx and metabolism, The authors’ good fit to the data from Sargent and Park is fortuitously due to the fact that all the bicarbonate generated from acetate is accounted for in their model by bicarbonate influx, inflating the dialysance of this anion. The reason for this concordance is simple to understand. To the extent that bicarbonate generated by acetate metabolism is added to the ECF fluid, it reduces the dialysate/blood concentration gradient and thereby reduces direct bicarbonate entry across the dialysis membrane reciprocally. Acetate dynamics are well studied and easily quantifiable in their model. Acetate’s role in acid-base balance is unique in that it provides a stable rate of bicarbonate addition throughout treatment, in contrast to the continually changing rate of direct bicarbonate influx. See Sargent et al, Seminars in Dialysis,2020: https://doi.org/10.1111/sdi.12902 2020;33:402-409, a study in which all the net alkali added is from acetate influx and metabolism.

2. Adding acetate influx and metabolism to the model will also lower DHCO3- and increase DCO2, correcting the authors’ unexplained calculation that DCO2 is much lower than DHCO3-. This finding is incompatible with the physical properties of the two substances. The membrane is freely permeable to CO2. Reflecting this physical property of CO2, the post membrane blood PCO2 is quite high, approaching its partial pressure in the bath solution.

3. A major problem of all modeling efforts when using clinical data for validation is that they have to “fit” their variables to the data to obtain a minimum LS for the difference between modeled and measured data. To do this they have to adjust variables in an iterative fashion This is true of the Sargent model as well as the authors’ model. In their model, they “tuned” the dialyances of bicarbonate and dissolved CO2: in the Sargent model, the variable mH+ was tuned. In the authors’ model, inclusion of acetate may well change the tuned values significantly. As noted above, I think it will increase the dialysance of CO2, which is inexplicably lower than that of bicarbonate ions in this paper.

4. I may be missing something, but I disagree with the authors’ conclusion that a hemoglobin effect in the blood leaving the dialysis membrane and increased ventilation occurring during the dialysis session can explain the failure of blood bicarbonate concentration to reach equilibrium with the bath fluid. Carbon dioxide added to the blood traversing the dialysis membrane and its reaction with hemoglobin actually increases afferent blood bicarbonate concentration notably, overwhelming any effect of titration of hemoglobin by added bicarbonate. See Gennari et al, Kidney Medicine (2022), doi: https://doi.org/10.1016/j.xkme.2022.100523. With regard to increased ventilation lowering PCO2, data in intact experimental animals and humans show that the acute effect on blood bicarbonate concentration is trivial – 0.1 mmols/L for each mmHg drop in PCO2, and even a tiny drop in blood bicarbonate concentration will increase the transmembrane concentration gradient resulting in more bicarbonate influx.

5. Two papers by the Sargent/Gennari group published this year are very relevant to the authors’ paper. The one in Kidney Medicine (cited above in comment 4j) reviews many of the issues discussed in this paper, including the enigma of the lack of continued increase in blood bicarbonate concentration despite the continued presence of a driving force for bicarbonate influx. The second demonstrates the power of the Sargent model. In that paper, in a patient study, bath bicarbonate concentration was increased in a stepwise fashion at timed intervals during hemodialysis. Analytic solution of the differential equations in the model accurately predicted the resultant blood bicarbonate concentration achieved at each stepwise increase in bath bicarbonate concentration. See Marano et al IEEE, 2022;10:17473-17483. 10.1109/ACCESS.2022.3147261.

6. On page 13 of the manuscript, the authors make the strange statement that it is often assumed that the distribution of bicarbonate and non-bicarbonate buffers is confined to the extracellular space. While bicarbonate ions themselves are largely confined to the ECF, it has been known for over fifty years, that the intracellular compartment continually interacts with and buffers ECF bicarbonate. This is the basis of the so called “bicarbonate space of distribution” which is as low as 40-50% of body water when blood bicarbonate is normal and rises to 80% in metabolic acidosis.

7. In the introduction, the authors state that in kidney failure metabolic acidosis can develop due to the inability of the kidneys to secrete sufficient bicarbonate anions. This is incorrect. The metabolic acidosis in kidney failure is due to the inability of the kidney to excrete the hydrogen ions produced by endogenous acid production.

8. I am confused by your use of the word “state”. Is it a synonym for steady state? Of does it indicate the state at the moment of measurement or calculation?

9. The term: V̇CO2,t – tissue CO2 consumption, is confusing. Surely you don’t mean that tissues consume CO2. We don’t have chlorophyll.

Reviewer #2: This submission describes a novel model of acid-base balance in dialysis patients. The model was shown to fit data collected during previously published studies. The model was adapted from previously published models. The novel features included the inclusion of lung function, dialyzer function, volume changes caused by ultrafiltration and transfer of CO2 from dialysate into blood.

I found the paper to provide valuable insight into the process which determine pH and bicarbonate concentrations during dialysis. The role of CO2 in the observed changes in serum bicarbonate levels was instructive and convincing.

My comments below

1) The simulations over the weekly cycle were not compared to measurements. I am wondering if additional buffers (e.g. in the bone space) may come into play as pH falls after long inter-dialytic interval. The introduction mentions bone-reabsorbing effects of acidosis, which would, presumably limit the fall in pH and bicarbonate.

2) The model ignores residual renal function. Regeneration of bicarbonate by the kidneys is likely to have a major impact on changes in pH and bicarbonate concentrations during the intervals between dialysis. The paper should comment on the potential impact of residual renal function.

3) minor point: page 22:"These simulations show how sensible the whole system is to changes in the respiration.." it should be sensitive, not sensible.

---

## [Author Response · Author response to Decision Letter 0]

16 Dec 2022

We thank the Editor and the Reviewers for the time spent to review are work, which we hope lead to an improved manuscript. Following the comments of the Reviewers, we modified our mathematical model and replaced the description of the methods, numerical results, and figures in the manuscript with updated version. The general results were largely unchanged, but we did significant additions to the discussion. Other editorial changes were performed for better readability. 

 In the following rebuttal, our answers will be directly presented below each comment.

Reviewer #1: The authors have adapted a 4-compartment (plasma, erythrocytes, interstitial fluid and intracellular fluid) model of acid-base regulation in individuals with normal kidney function to analyze the acid base response to rapid alkalinization during hemodialysis. In contradistinction to earlier models, they also include the effect of respiratory control over the partial pressures of CO2 and oxygen. They have evaluated their model using data obtained in 2 patient studies in which blood bicarbonate concentration, pH and PCO2 were measured at timed intervals during treatment. They conclude that their model fits the clinical measurements well and assert that they have uncovered fresh insights into the complex acid-base interactions during hemodialysis. My comments are below.

1. A significant flaw in their model is to ignore the role of acetate influx and metabolism in bicarbonate addition to the patient during hemodialysis. They justify their lack of inclusion of by citing evidence that variations in bath acetate have no effect on the end-dialysis acid-base status in patients receiving hemodialysis. While this is true, it doesn’t mean that acetate influx and metabolism plays no role in contributing bicarbonate to the ECF fluid. Sargent et al have shown that 1/3 of the net bicarbonate added is from acetate influx and metabolism, The authors’ good fit to the data from Sargent and Park is fortuitously due to the fact that all the bicarbonate generated from acetate is accounted for in their model by bicarbonate influx, inflating the dialysance of this anion. The reason for this concordance is simple to understand. To the extent that bicarbonate generated by acetate metabolism is added to the ECF fluid, it reduces the dialysate/blood concentration gradient and thereby reduces direct bicarbonate entry across the dialysis membrane reciprocally. Acetate dynamics are well studied and easily quantifiable in their model. Acetate’s role in acid-base balance is unique in that it provides a stable rate of bicarbonate addition throughout treatment, in contrast to the continually changing rate of direct bicarbonate influx. See Sargent et al, Seminars in Dialysis,2020: https://doi.org/10.1111/sdi.12902 2020;33:402-409, a study in which all the net alkali added is from acetate influx and metabolism.

Answer: We appreciate the Reviewer’s comments regarding the importance of acetate transfer into the patient. Although we did not include acetate in our initial model because it was not present in the physiological acid-base model of Rees and Andreassen, after reviewing the seminal work of Sargent et al (2018, 2019, 2020), we now agree that our initial model was incomplete without including acetate kinetics in the model. As suggested by the reviewer, the addition of acetate kinetics could be easily quantified in a revised model by mean of direct calculation of the flow of bicarbonate generated by conversion of acetate in blood, implemented in Equation 1 of our revised manuscript, as expressed by equation A3 in the 2018 paper by Sargent et al (Seminars in Dialysis, 2018, DOI: 10.1111/sdi.12714). For simplicity, because the formulation of the Rees model makes it difficult to implement the same ODEs for acetate without substantial modifications to Rees’s system of biochemistry equations, we assumed a constant blood acetate concentration in our simulations. We believe this is a reasonable assumption given the observation in the same Sargent paper of the speed with which acetate flow reached a steady state value. Anyway, simulations of Sargent’s data using a time-dependent acetate concentration calculated with equation A1 in their paper brought similar results to the constant concentration assumption.

The addition of acetate to our revised model brought several changes to the results. First, as the Reviewer predicted, the estimated dialysance for bicarbonate (DHCO3-) was lower in presence of acetate (125 ml/min vs. previous 190 ml/min for Sargent’s data), while the estimated dialysance for CO2 (DCO2) was mainly affected by the changes described in our answer to the next comment.

2. Adding acetate influx and metabolism to the model will also lower DHCO3- and increase DCO2, correcting the authors’ unexplained calculation that DCO2 is much lower than DHCO3-. This finding is incompatible with the physical properties of the two substances. The membrane is freely permeable to CO2. Reflecting this physical property of CO2, the post membrane blood PCO2 is quite high, approaching its partial pressure in the bath solution.

Answer: concerning the low estimated value of CO2 dialysance, after careful consideration of the Reviewer’s comment we went back to look at one the (unfortunately) few instances of published data that allow for the calculation of DCO2, Sombolos’ 2005 paper in Artificial Organs (29(11):892-8. doi: 10.1111/j.1525-1594.2005.00126.x). It reports a DCO2 calculated on dialysate side, with a dialysate flow rate (Qd) of 0.7 L/min, equal to 0.448 L/min. Calculating the same dialysance from blood concentrations, with Qb = 0.3 L/min results in a dialysance of 0.177 L/min. Assuming 35% hematocrit (unfortunately a measured value was missing from the data), the same dialysance calculated with plasma water flow rate (instead of Qb) equals 0.107 L/min, much closer to the value estimated by our model in the first submitted version. Similar results are obtained for bicarbonate, with a reported dialysate-side DHCO3- = 0.133 L/min. Calculating it on blood side it results in DHCO3- = 0.223 L/min; DHCO3- for plasma water flow rate equals 0.135 L/min. 

We found only one other study, published by Symreng et al (Kidney International, Vol. 41 (1992), pp. 1064—1069, doi.org/10.1038/ki.1992.162), that reported afferent and efferent concentrations of PCO2 and bicarbonate in dialysate and blood across the hemodialyzer. The blood and dialysate-side dialysances for bicarbonate and CO2 had similar relative magnitudes to those obtained from the Sombolos’ data (again we assumed 35% hematocrit to calculate plasma flow rate). The main difference was that, with Symreng’s data, the dialysate-side DHCO3- was fairly different from DHCO3- calculated with plasma water flow (we assume that the almost identical values obtained in Sombolos’ data are a coincidence).

The dialysance for CO2 previously estimated by our model was a parameter used exclusively in the Equation 2 of the manuscript, which described the increase only in plasma dissolved CO2 concentration caused by dialysis, regardless of the effective distribution volume of CO2. The increase in plasma CO2 was then propagated to changes in other variables in plasma and RBC when solving the system of equations for post-dialyzer full blood; this derived from the way that the Rees and Andreassen model was originally formulated. We believe this is also the reason why the estimated value of DCO2 was similar to the value calculated from the above-described clinical data using plasma flow rates. Anyway, in this revised version of the model, we changed Equation 2 to instead describe the increase in pCO2 as a result of dialysis, similarly to how we did in a previous paper (BBE 2021, doi.org/10.1016/j.bbe.2021.07.006):

 〖pCO〗_"2,out" =〖pCO〗_(2,a)+D_CO2/(0.93⋅Q ˙_b ) (〖pCO〗_(2,d)-〖pCO〗_(2,a) ) (2)

Using pCO2 instead of dissolved CO2 it was possible to write Equation 2 for full blood instead of just plasma, which resulted in two improvements. First, we can more correctly use in the equation the blood flow rate for CO2, reflecting the ease with which CO2 molecules cross RBC barriers. Second, the estimated values of DCO2 are now higher and closer to the clinical values reported above and obtained from calculating blood-side dialysances using blood flow rate. However, as we point out in the revised manuscript, because of the inconsistency of dialysate-side and blood-side calculation of dialysance from concentration data, we still suggest that the dialysance estimated by our model might represent a slightly different parameter and perhaps should not be directly compared.

Anyway, our estimated dialysances result, for Sargent data, in concentrations at the inlet and outlet of the dialyzer blood channel which are similar to what reported by Sombolos or Symreng:

 pCO2_in (mmHg) pCO2_out (mmHg) HCO3-_in (mmol/L) HCO3-_out (mmol/L)

Model 37.47 53.33 22.10 28.88

Sombolos 38.3 62.8 20.4 29.4

Symreng 39.8 50.4 29.0 32.2

Of course, the data and our results are not completely comparable, as there are differences in the dialysate composition and dialysis flow rates between these studies, but we believe they at least show that the dialysances estimated by our model are not outside of the feasible range.

As a final note regarding the expectation that the dialysance of a CO2, a molecule smaller than HCO3, should have a larger dialyzer than HCO3, the previous publication by Sombolos et al reports a dialysance for O2, a molecule smaller than HCO3, of 0.119 L/min, a value that is smaller than reported for HCO3 of 0.134 L/min. Thus, it should not be assumed that smaller gas molecules will traverse a hemodialysis membrane faster than an anion, such as HCO3; this may be a unique physical chemical property of the specific hemodialysis membranes used in these studies.

3. A major problem of all modeling efforts when using clinical data for validation is that they have to “fit” their variables to the data to obtain a minimum LS for the difference between modeled and measured data. To do this they have to adjust variables in an iterative fashion This is true of the Sargent model as well as the authors’ model. In their model, they “tuned” the dialysances of bicarbonate and dissolved CO2: in the Sargent model, the variable mH+ was tuned. In the authors’ model, inclusion of acetate may well change the tuned values significantly. As noted above, I think it will increase the dialysance of CO2, which is inexplicably lower than that of bicarbonate ions in this paper.

Answer: A response to this comment can already be found in the above answers.

4. I may be missing something, but I disagree with the authors’ conclusion that a hemoglobin effect in the blood leaving the dialysis membrane and increased ventilation occurring during the dialysis session can explain the failure of blood bicarbonate concentration to reach equilibrium with the bath fluid. Carbon dioxide added to the blood traversing the dialysis membrane and its reaction with hemoglobin actually increases afferent blood bicarbonate concentration notably, overwhelming any effect of titration of hemoglobin by added bicarbonate. See Gennari et al, Kidney Medicine (2022), doi: https://doi.org/10.1016/j.xkme.2022.100523. With regard to increased ventilation lowering PCO2, data in intact experimental animals and humans show that the acute effect on blood bicarbonate concentration is trivial – 0.1 mmols/L for each mmHg drop in PCO2, and even a tiny drop in blood bicarbonate concentration will increase the transmembrane concentration gradient resulting in more bicarbonate influx.

Answer: we appreciate the Reviewer’s criticism. We based our explanation for the model’s behavior regarding DCO2 and the dampening of bicarbonate curve from the papers published by Marano et al (World J Nephrol 2016 and Blood Purif 2016) about dialysate-induced acidosis. Simulations that we performed with DBic = 0 and only dissolved CO2 transport (not shown in the manuscript) showed that plasma pH and bicarbonate levels decrease during dialysis, for effect of the increase in pCO2 (even when assuming zero net endogenous acid generation), effectively inducing acidosis in the patient. Conversely, we show in the Figure 6 of the manuscript that bicarbonate increases less in the presence of dissolved CO2 transport, and at the same time pCO2 decreases when no CO2 transport is assumed. Our interpretation was thus that the concomitant increase in plasma bicarbonate and pCO2 results in dampened increase in plasma bicarbonate and pH overall. The latter is a direct result of the Henderson-Hasselbalch equation: increasing bicarbonate concentration in a solution alone by a certain amount, the resulting increase in pH will be higher than if we increase both bicarbonate and CO2. The equilibrium between plasma and erythrocyte bicarbonate, CO2, pH, and erythrocyte hemoglobin forms in the Rees model is regulated by a series of H-H equations, therefore suggesting that is the logic behind the observed simulated results. We did not modify the blood chemistry formulation of the Rees model in adapting our model to HD, so we understand that we are implicating trusting the original formulation which, however validated, could be not completely suited to describe the chemistry of uremic patients. 

Unfortunately, we came to realize that the specific formulation of the Rees model makes it difficult to give interpretations to the results in terms of transport of H+ ions, because for most of the simulated quantities, their changes in time are given by a series of steady-states that are constantly recalculated. Therefore, we cannot directly derive insights on the source of the H+ ions titrating the added bicarbonate using are model, just observe the effects of the treatment DCO2 setting on bicarbonate concentration. We cannot therefore make guesses on the source of the H+ ions titrating the added bicarbonate based on our simulations. We added these comments in the Discussion of the revised manuscript:

“It must be noted that, because of the formulation of the original model of Rees and Andreassen, changes in quantities such as bicarbonate, pH (H+ concentration), and dissolved CO2 in plasma and erythrocytes are not calculated as result of modelling of their transport flows, but rather as solutions of a series of steady-states for the chemistry equations of each blood and fluid compartment. As such, it is difficult to directly assess from our modeling results the origin of the H+ anions titrating the dialyzed bicarbonate.”

Regarding the second part of the Reviewer’s comment, considering the effect of ventilation increase on bicarbonate, we believe that the most direct impact of increasing the respiration rate is simply allowing for more transport of CO2 to blood without increasing plasma pCO2 (and thus still fitting the data), and it is this CO2 addition that affect bicarbonate plasma levels.

5. Two papers by the Sargent/Gennari group published this year are very relevant to the authors’ paper. The one in Kidney Medicine (cited above in comment 4j) reviews many of the issues discussed in this paper, including the enigma of the lack of continued increase in blood bicarbonate concentration despite the continued presence of a driving force for bicarbonate influx. The second demonstrates the power of the Sargent model. In that paper, in a patient study, bath bicarbonate concentration was increased in a stepwise fashion at timed intervals during hemodialysis. Analytic solution of the differential equations in the model accurately predicted the resultant blood bicarbonate concentration achieved at each stepwise increase in bath bicarbonate concentration. See Marano et al IEEE, 2022;10:17473-17483. 10.1109/ACCESS.2022.3147261.

Answer: the two new papers suggested by the Reviewer were added to the discussion.

6. On page 13 of the manuscript, the authors make the strange statement that it is often assumed that the distribution of bicarbonate and non-bicarbonate buffers is confined to the extracellular space. While bicarbonate ions themselves are largely confined to the ECF, it has been known for over fifty years, that the intracellular compartment continually interacts with and buffers ECF bicarbonate. This is the basis of the so called “bicarbonate space of distribution” which is as low as 40-50% of body water when blood bicarbonate is normal and rises to 80% in metabolic acidosis.

Answer: it was a wrong statement at it has been removed.

7. In the introduction, the authors state that in kidney failure metabolic acidosis can develop due to the inability of the kidneys to secrete sufficient bicarbonate anions. This is incorrect. The metabolic acidosis in kidney failure is due to the inability of the kidney to excrete the hydrogen ions produced by endogenous acid production.

Answer: thank you for the correction, we rewrote that sentence.

8. I am confused by your use of the word “state”. Is it a synonym for steady state? Of does it indicate the state at the moment of measurement or calculation?

Answer: when in the manuscript we wrote of state variable, or state equations, we meant the state-space variables that determine the state (solution at a time t) of a system of ordinary differential equations.

9. The term: V̇CO2,t – tissue CO2 consumption, is confusing. Surely you don’t mean that tissues consume CO2. We don’t have chlorophyll.

Answer: thank you for spotting this typo, it was corrected.

Reviewer #2: This submission describes a novel model of acid-base balance in dialysis patients. The model was shown to fit data collected during previously published studies. The model was adapted from previously published models. The novel features included the inclusion of lung function, dialyzer function, volume changes caused by ultrafiltration and transfer of CO2 from dialysate into blood.

I found the paper to provide valuable insight into the process which determine pH and bicarbonate concentrations during dialysis. The role of CO2 in the observed changes in serum bicarbonate levels was instructive and convincing.

My comments below

1) The simulations over the weekly cycle were not compared to measurements. I am wondering if additional buffers (e.g. in the bone space) may come into play as pH falls after long inter-dialytic interval. The introduction mentions bone-reabsorbing effects of acidosis, which would, presumably limit the fall in pH and bicarbonate.

Answer: it is indeed possible that over the course of several days these factors would come into play. They are not currently accounted for by our model, but they’re effect could be perhaps included in the choice of net acid generation rate. This parameter was in general assumed based on reported values (1 mmol/day/Kg), and only in the simulation of the weekly cycle it was tuned to provide cyclic steady-state.

2) The model ignores residual renal function. Regeneration of bicarbonate by the kidneys is likely to have a major impact on changes in pH and bicarbonate concentrations during the intervals between dialysis. The paper should comment on the potential impact of residual renal function.

Answer: we thank the reviewer for this comment. Residual renal function was neglected for simplicity, but a sentence on this issue was added to the discussion.

3) minor point: page 22:"These simulations show how sensible the whole system is to changes in the respiration." it should be sensitive, not sensible.

Answer: we thank the reviewer for spotting this typo.

---

## [Decision Letter · Decision Letter 1]

8 Feb 2023

Mathematical modelling of bicarbonate supplementation and acid-base chemistry in kidney failure patients on hemodialysis

PONE-D-22-15261R1

Dear Dr. 

We’re pleased to inform you that your manuscript has been judged scientifically suitable for publication and will be formally accepted for publication once it meets all outstanding technical requirements.

Kind regards,

Emre Avci

Academic Editor

PLOS ONE

---

## [Editor Report · Acceptance letter]

15 Feb 2023

PONE-D-22-15261R1 

Mathematical modelling of bicarbonate supplementation and acid-base chemistry in kidney failure patients on hemodialysis 

Dear Dr. Pietribiasi:

I'm pleased to inform you that your manuscript has been deemed suitable for publication in PLOS ONE. Congratulations! Your manuscript is now with our production department. 

Kind regards, 

on behalf of

Dr. Emre Avci 

Academic Editor

PLOS ONE